

# Glomerular filtration rate correlation and agreement between common predictive equations and standard 24-hour urinary creatinine clearance in medical critically ill patients

Suwikran Wongpraphairot[1,*], Attamon Thongrueang[2] and Rungsun Bhurayanontachai[3,*]

[1] Nephrology Unit, Department of Internal Medicine, Faculty of Medicine, Prince of Songkla University, Hat Yai, Songkhla, Thailand
[2] Department of Internal Medicine, Faculty of Medicine, Prince of Songkla University, Hat Yai, Songkhla, Thailand
[3] Critical Care Medicine Unit, Department of Internal Medicine, Faculty of Medicine, Prince of Songkla University, Hat Yai, Songkhla, Thailand
[*] These authors contributed equally to this work.

Corresponding author
Rungsun Bhurayanontachai,
rungsun2346@gmail.com

## ABSTRACT

**Background**. Determining kidney function in critically ill patients is paramount for the dose adjustment of several medications. When assessing kidney function, the glomerular filtration rate (GFR) is generally estimated either by calculating urine creatinine clearance (UCrCl) or using a predictive equation. Unfortunately, all predictive equations have been derived for medical outpatients. Therefore, the validity of predictive equations is of concern when compared with that of the UCrCl method, particularly in medical critically ill patients. Therefore, we conducted this study to assess the agreement of the estimated GFR (eGFR) using common predictive equations and UCrCl in medical critical care setting.

**Methods**. This was the secondary analysis of a nutrition therapy study. Urine was collected from participating patients over 24 h for urine creatinine, urine nitrogen, urine volume, and serum creatinine measurements on days 1, 3, 5, and 14 of the study. Subsequently, we calculated UCrCl and eGFR using four predictive equations, the Cockcroft–Gault (CG) formula, the four and six-variable Modification of Diet in Renal Disease Study (MDRD-4 and MDRD-6) equations, and the Chronic Kidney Disease Epidemiology Collaboration (CKD-EPI) equation. The correlation and agreement between eGFR and UCrCl were determined using the Spearman rank correlation coefficient and Bland–Altman plot with multiple measurements per subject, respectively. The performance of each predictive equation for estimating GFR was reported as bias, precision, and absolute percentage error (APE).

**Results**. A total of 49 patients with 170 urine samples were included in the final analysis. Of 49 patients, the median age was 74 (21–92) years-old and 49% was male. All patients were hemodynamically stable with mean arterial blood pressure of 82 (65–108) mmHg. Baseline serum creatinine was 0.93 (0.3–4.84) mg/dL and baseline UCrCl was 46.69 (3.40–165.53) mL/min. The eGFR from all the predictive equations showed modest correlation with UCrCl (r: 0.692 to 0.759). However, the performance of all

the predictive equations in estimating GFR compared to that of UCrCl was poor, demonstrating bias ranged from −8.36 to −31.95 mL/min, precision ranged from 92.02 to 166.43 mL/min, and an unacceptable APE (23.01% to 47.18%). Nevertheless, the CG formula showed the best performance in estimating GFR, with a small bias (−2.30 (−9.46 to 4.86) mL/min) and an acceptable APE (14.72% (10.87% to 23.80%)), especially in patients with normal UCrCl.

**Conclusion**. From our finding, CG formula was the best eGFR formula in the medical critically ill patients, which demonstrated the least bias and acceptable APE, especially in normal UCrCl patients. However, the predictive equation commonly used to estimate GFR in critically ill patients must be cautiously applied due to its large bias, wide precision, and unacceptable error, particularly in renal function impairment.

## INTRODUCTION

In critically ill patients, dose adjustment of several medications depends on kidney function. Inappropriate dose administration may affect the efficacy and safety of drugs. The serum creatinine (SCr) level is not a good predictor of kidney function (*Seller-Perez et al., 2013*). Although serum cystatin c (SCys) was better correlated than SCr to identify acute kidney dysfunction (*Villa et al., 2005*), but the application of this specific type of non-glycosylated protein was limited in the critically ill patients due to the influence of inflammatory response on cystatin c production (*Finney, Newman & Price, 2000*). The glomerular filtration rate (GFR) is then recommended for estimating kidney function in acutely ill patients requiring drug dosage adjustment (*Andrassy, 2013*). Theoretically, GFR should be determined using a substrate that is fully filtered by the kidney without any tubular secretion or reabsorption. Exogenous substances, such as inulin, are strongly recommended as the gold standard for GFR measurement; however, in practice, the use of inulin is limited by cost and unavailability (*Seller-Perez et al., 2013*). Urinary creatinine clearance (UCrCl) has been proposed as an alternative method for GFR measurement, but it requires 24 h of urine collection and additional time for the test results to be reported. Therefore, this current standard method may not be suitable for general clinical practice. Several predictive equations of estimated GFR (eGFR) have been introduced, such as the Cockcroft–Gault (CG) formula, the four and six-variable Modification of Diet in Renal Disease Study (MDRD-4 and MDRD-6) equations, and the Chronic Kidney Disease and Epidemiology Collaboration (CKD-EPI) equation. These equations are calculated using serum creatinine levels and several anthropometric and demographic parameters, including age, sex, race, and body weight (*Sunder et al., 2014*). Nevertheless, there are several reports of augmented renal clearance (ARC) conditions in many types of critically ill patients, particularly those who are young and have experienced trauma, which may influence the performance of the predictive equations. ARC is generally defined as UCrCl >130 mL/min, which may enhance drug elimination and disturb the blood levels of several medications excreted

via renal clearance (*Kawano et al., 2016*; *Johnston et al., 2021*; *Baptista et al., 2020*; *Bilbao-Meseguer et al., 2018*). Then, there are concerns regarding the utilization of eGFR for drug dose adjustment in both overestimation and underestimation of actual GFR, which may influence the treatment outcomes. Although there were several previous studies regarding the performance on predictive equations, but the results remained inconsistent due to the difference kinds of patients and the incidence of ARC (*Al-Dorzi et al., 2021*; *Bouchard et al., 2009*; *da Silva Selistre et al., 2019*; *Tsai et al., 2018*). As there are limited data on the performance of predictive equations in critically ill patients, these equations may not offer an accurate evaluation of GFR in a critical care setting. In addition, most of the predictive equations generated from the general medical patients (*Sunder et al., 2014*), but the result of the performance of those equations in the medical critically ill patients was limited. Therefore, this study aimed to determine the correlation and agreement between the four predictive equations commonly used for estimating GFR and the standard UCrCl method and to determine the rate of ARC in a medical critical care setting.

## MATERIALS & METHODS

### Ethics

This is the secondary analysis of an ongoing nutrition therapy study comparing a peptide-based formula with a standard formula (TCTR20220221006; https://www.thaiclinicaltrials.org/show/TCTR20220221006. The original study and the secondary analysis were approved by the human research ethics committee (HREC) at the Faculty of Medicine, Prince of Songkla University (registration numbers REC.61-124-14-1 and REC.64-416-14-4, respectively). A written consent form approved by HREC and signed and dated by the patient's legally authorized representative or the patient at the time of consent.

### Study location

This study was conducted in the medical intensive care unit (ICU) and the respiratory care unit (RCU) at the Songklanagarind Hospital, where is the affiliated hospital of Faculty of Medicine, Prince of Songkla University, Thailand.

### Study population

The mechanically-ventilated adult patients, who were admitted the medical ICU and RCU between February 2019 to March 2022, were screened. The inclusion criteria were as follows: (1) modified Nutrition Risk in the Critically Ill (NUTRIC) score $\geq$5; (2) body mass index, 18–30 kg/m$^2$; (3) receiving enteral nutrition within 48 h after admission; and (4) hemodynamically stable with a low dose of norepinephrine <0.3 mcg/kg/min. We excluded patients with (a) a risk of aspiration; (b) thyroid disorder; (c) severe hepatic impairment; (d) renal replacement therapy requirement; (e) abdominal hypertension; (f) autoimmune disease; (g) human immunodeficiency virus infection; (h) with immunosuppressive therapy; and (i) terminal illness. The recruited patients received either the study formula or the control formula via a gastric feeding tube for 14 days.

**Table 1** Predictive equations for the estimated glomerular filtration rate (GFR).

| Predictive equations | Calculation details |
|---|---|
| Cockcroft—Gault (CG) formula | $GFR = ([140 - Age(years)] \times Weight(kg))/(7.2 \times SCr(mg/dL)) \times 0.85(if\ female)$ |
| Four-variable Modification of Diet In Renal Disease Study (MDRD4) equation | $GFR = 175 \times [SCr(\frac{mg}{dL})]^{-1.154} \times [Age\ (years)]^{-0.203} \times 0.742(if\ female) \times 1.212(if\ African\ American)$ |
| Six-variable Modification of Diet In Renal Disease Study (MDRD6) equation | $GFR = 161.5 \times [SCr(\frac{mg}{dL})]^{-0.999} \times [Age\ (years)]^{-0.176} \times 0.762(if\ female) \times 1.18(if\ African\ American) \times [Serum\ BUN(\frac{mg}{dL})]^{-0.17} \times [Serum\ Albumin(\frac{g}{dL})]^{+0.318}$ |
| Chronic Kidney Disease Epidemiology Collaboration (CKD-EPI) formula | $GFR\ Female = 144 \times [\frac{SCr(\frac{mg}{dL})}{0.7}]^{-1.209} \times 0.993^{Age\ (years)}$ <br> $GFR\ Male = 141 \times [\frac{SCr(\frac{mg}{dL})}{0.9}]^{-1.209} \times 0.993^{Age\ (years)}$ |

**Notes.**

BUN, blood urea nitrogen; SCr, serum creatinine.

## Data collection

From all variables of the primary study, we extracted the data regarding baseline demographic and clinical data including age, sex, cause of admission, Acute Physiology and Chronic Health Evaluation (APACHE II) score, Sequential Organ Failure Assessment (SOFA) score, need for inotropes or vasopressors, body weight, height, biochemistry results, and daily caloric and protein supplement intake.

During the 14-day study period, serum samples were collected for serum albumin, serum blood urea nitrogen (BUN), and serum creatinine (SCr) measurements, and urine was collected over 24 h for urine nitrogen, urine creatinine (UCr), and urine volume measurements on days 1, 3, 5, and 14. Age, body weight, sex, and BUN and serum albumin levels were collected on the same day of urine collection to determine the eGFR using predictive equations. SCr and UCr were analyzed by the enzymatic method using Abbott Alinity c platform.

## Determination of kidney function

Standard UCrCl values were calculated using the UCr and SCr values and urine volume for 24 h by using the standard method: (UCr × urine flow in mL/min)/SCr. Four predictive equations, the Cockcroft–Gault (CG) formula, the four and six-variable Modification of Diet in Renal Disease Study (MDRD-4 and MDRD-6) equations, and the Chronic Kidney Disease and Epidemiology Collaboration (CKD-EPI) equation were used to calculate eGFR, as described in Table 1.

## Operational definitions

ARC was identified if the UCrCl was > 130 mL/min. According to UCrCl, we classified the patients into two groups: low-UCrCl group and normal-UCrCl group. The low-UCrCl group was defined when UCrCl < 60 mL/min and the normal-UCrCl group was indicated when UCrCl ≥ 60 mL/min. The cutoff value of UCrCl at 60 mL/min was extrapolated from the recommendations for dose adjustment for drug administration in patients with kidney impairment (*Ahern & Possidente, 2013*).

## Statistical analysis

The sample size of the study was derived from the results of a previous study (*Baptista et al., 2011*), in which the difference in GFR when using UCrCl and the CG formula was

19.9 ± 76.8 mL/min. For the Bland–Altman method with a type-I and type-II error of 0.1 and 0.2, respectively, a total of 52 pairs of UCrCl and eGFR from predictive equations were eventually required. The sample size estimation method for Bland-Altman method was applied according to the current recommendation (*Lu et al., 2016*).

The Kolmogorov–Smirnov test was used to assess the normal distribution of continuous variables. Continuous data are expressed as mean and standard deviation (SD) or median and minimum-maximum values, depending on the data distribution. Categorical variables are expressed as numbers and percentages. No missing data were corrected.

The incidence of ARC was reported as numbers and percentages. The Spearman correlation coefficient (r) was used to determine the level of correlation between UCrCl and eGFR using a predictive equation and was presented as a correlogram. The level of correlation was classified using the standard recommendation (*Astivia & Zumbo, 2017*).

The agreement between UCrCl and GFR from the four predictive equations was assessed using a Bland–Altman plot with multiple measurements per subject. The bias and precision were reported as the mean difference between UCrCl and eGFR from predictive equations and its 95% confidence interval (CI) and the upper and lower limits of agreement of mean difference (mean difference ± 1.96 SD) and its 95% CI, respectively. The absolute percentage error (APE) between the methods is also reported. The APE was calculated as $100 \times$ (eGRF from each predictive equation - UCrCl)/UCrCl. An acceptable APE between the methods was defined as < 30% (*Critchley & Critchley, 1999*).

We also performed subgroup analysis to determine the agreement between UCrCl and eGFR from predictive equations between the low-UCrCl group and normal UCrCl group. The report of agreement including bias and precision between UCrCl and eGFR in each group was described as above.

The sample size estimation and all statistical analyses were performed using MedCalc® Statistical Software version 20.022 (MedCalc Software Ltd., Ostend, Belgium; https://www.medcalc.org; 2021). Statistical significance was set at $P < 0.05$. We consider the STROBE statement (STrengthening the Reporting of OBservational studies in Epidemiology) (*Von Elm et al., 2008*) and make sure that our report follows the standards for reporting observational studies outlined.

## RESULTS

### Patient characteristics

Finally, 49 patients were included in this study. All the clinical data are described in Table 2. Of the 49 patients, 24 (49%) were men, with a median age of 74 (21–92) years. The median APACHE II and SOFA scores were 27 (17–42) and 6 (2–18), respectively. The main cause of admission was respiratory in nature (28 patients [58.1%]). The estimated body weight was 58 (38–90) kg. The mean daily calories and protein supplementation were 1,265.15 (267.1) kcal/day and 63.26 (13.36) g/day. The median BUN and SCr levels at baseline were 22.6 (2.3–96.6) mg/dL and 0.93 (0.3–4.84) mg/dL, respectively. All patients were hemodynamically stable with the mean arterial blood pressure of 82 (65–108) mmHg.

Regarding 24-hour urine collection, complete numbers of urine collection was performed in 63.3% (31/49) patients, and in 6.1% (3/49), only one urine sample was

**Table 2  Patient clinical characteristics.** All continuous variables are presented as medians with minimum and maximum values.

| Clinical Characteristics | Results |
|---|---|
| Age (years) | 74 (21–92) |
| Male, N (%) | 24 (49%) |
| APACHE II score | 27 (17–42) |
| SOFA score | 6 (2–18) |
| Cause of admission, N (%) | |
| • Respiratory cause | 28 (57.1%) |
| • Cardiovascular cause | 12 (24.5%) |
| • Neurological cause | 4 (8.2%) |
| • Infectious cause | 2 (4.1%) |
| • Sepsis/septic shock | 2 (4.1%) |
| • Hepatobiliary cause | 1 (2.0%) |
| Systolic blood pressure (mmHg) | 122 (95-163) |
| Diastolic blood pressure (mmHg) | 62 (37-89) |
| Mean arterial blood pressure (mmHg) | 82 (65-108) |
| Inotrope/vasopressor requirement, N (%) | 1 (2.0%) |
| Estimated body weight (kg) | 58 (37–90) |
| Average energy supplementation (kcal/day) | 1,272.86 (679–1,916.93) |
| Average protein supplementation (g/day) | 63.64 (33.95–95.85) |
| Average energy supplementation by weight (kcal/kg/day) | 23.27 (9.83–31.95) |
| Average protein supplementation by weight (g/kg/day) | 1.16 (0.49–1.60) |
| Baseline serum BUN (mg/dL) | 22.6 (2.3–96.6) |
| Baseline serum Cr (mg/dL) | 0.93 (0.3–4.84) |
| Baseline urine volume (mL/day) | 1,250 (330-3,700) |
| Baseline Urine creatinine clearance (mL/min) | 46.69 (3.40-165.53) |
| Baseline serum albumin (g/dL) | 2.73 (1.07-4.78) |

**Notes.**
APACHE II, Acute Physiology and Chronic Health Evaluation Score; SOFA, Simplified Organ Failure Assessment Score; BUN, blood urea nitrogen; Cr, creatinine.

collected during the study period. The baseline urine volume was 1,250 (330–3,700) mL/day. Finally, 170 pairs of eGFR and UCrCl values were obtained from our cohort for the agreement analysis.

The mean values of UCrCl and eGFR measured using CG, CKD-EPI, MDRD-4, and MDRD-6 were $57.58 \pm 32.26$ mL/min, $66.45 \pm 37.93$ mL/min, $77.29 \pm 31.18$ mL/min, $91.47 \pm 57.17$ mL/min, and $79.69 \pm 47.80$ mL/min, respectively. The UCrCl was significantly lower than the eGFR for all the predictive equations ($P < 0.001$).

## Correlation between UCrCl and eGFR from predictive equations

We performed a correlation analysis for UCrCl and eGFR from four commonly used predictive equations using the Spearman rank correlation coefficient. All eGFR values from the predictive equations showed modest correlation. Nevertheless, the MDRD-6 equation had the strongest correlation with the standard UCrCl method, with a correlation coefficient of 0.76, followed by the MDRD-4 formula ($r = 0.73$), the CKD-EPI formula

**Table 3  Correlograms of UCrCl and estimated GFR using the four predictive equations.**

|         | MDRD4 | CKD-EPI | MDRD6 | CG    | UCrCl |
|---------|-------|---------|-------|-------|-------|
| **MDRD4**   |       | 0.968   | 0.973 | 0.886 | 0.723 |
| **CKD-EPI** | 0.968 |         | 0.937 | 0.915 | 0.731 |
| **MDRD6**   | 0.973 | 0.937   |       | 0.870 | 0.759 |
| **CG**      | 0.886 | 0.915   | 0.870 |       | 0.692 |
| **UCrCl**   | 0.723 | 0.731   | 0.759 | 0.692 |       |

**Notes.**

The correlation was determined using the Spearman rank correlation coefficient. MDRD4, four-variable Modification of Diet in Renal Disease Study equation; MDRD6, six-variable Modification of Diet in Renal Disease Study equation; CKD-EPI, Chronic Kidney Disease Epidemiology Collaboration equation; CG, Cockroft–Gault formula; UCrCl, urinary creatinine clearance.

($r = 0.72$), and the CG formula ($r = 0.69$). The correlation between UCrCl and eGFR from predictive equations is illustrated in Table 3.

## Agreement between UCrCl and eGFR from predictive equations

From the Bland–Altman plot with multiple measurements per subject of the 170 urine samples, we found that all the eGFR values from the four predictive equations were significantly higher than the UCrCl value. In addition, the precision of some predictive equations were found to be wide, and the APE was unacceptable, particularly for the two MDRD equations. The MDRD-4 formula had bias of $-31.45$ ($-40.53$ to $-27.26$) mL/min, precision 166.43 mL/min with APE of 47.18% (37.21% to 64.84%). The MDRD-6 formula had bias $-20.68$ ($-27.30$ to $-16.93$) mL/min, precision 129.92 mL/min and APE 33.78% (27.53% to 39.90%). In addition, CKD-EPI formula demonstrated bias $-18.96$ ($-23.25$ to $-16.18$) mL/min, precision 92.02 mL/min and APE 29.62% (23,99% to 45.38%). However, the CG formula had the lowest bias ($-8.36$ ml/min) and APE <30% (Table 4). The Bland–Altman plot with multiple measurements per subject for UCrCl and eGFR from the four predictive equations is illustrated in Fig. 1.

In our subgroup analyses, we found that 98 (57.6%) urine samples were in the low-UCrCl group (Table 5). The bias and precision of the CG and CKD-EPI formulas became larger and wider in the low-UCrCl group, respectively, but both the MDRD equations remained unchanged. In low-UCrCl group, bias of CG amd CKD-EPI formula was $-13.69$ ($-19.53$ to $-7.85$) mL/min and $-26.59$ ($-31.35$ to $-21.83$) mL/min, respectively. The precision of CG and CKD-EPI formula was 114.13 mL/min and 93.04 mL/min, respectively. All equations had unacceptable APE ranged between 29.57% to 73.26%. In the normal-UCrCl group, the CG formula demonstrated the better performance than the other formula. The eGFR from the CG formula had the lowest bias ($-2.30$; ($-9.46$, 4.86) mL/min), which was not significantly different ($P = 0.52$) from the UCrCl method. Although the precision was still high, the APE was acceptable.

## Incidence of ARC

Only four urine samples from two patients out of 170 samples (2.4%) had a UCrCl value greater than 130 mL/min/m$^2$. One patient was middle-aged men who presented with acute

**Table 4   Bias, precision, and absolute percentage error of estimated glomerular filtration rate using predictive equations versus urinary creatinine clearance (N = 170).**

| Pair agreement | Bias (mL/min) [95% CI] | Precision | | | Absolute percentage error (%) [95% CI] |
|---|---|---|---|---|---|
| | | Upper limit of agreement (mL/min) [95% CI] | Lower limit of agreement (mL/min) [95% CI] | Width of limit of agreement (mL/min) | |
| CG and UCrCl | −8.36[*] [−13.43 to −4.30] | 50.28 [39.36 to 64.63] | −67.0 [−81.35 to −56.08] | 117.28 | 23.01% [19.81 to 29.52] |
| CKD-EPI and UCrCl | −18.96[*] [−23.25 to −16.18] | 27.05 [19.10 to 37.48] | −64.97 [−75.39 to −57.01] | 92.02 | 29.62% [23.99 to 45.38] |
| MDRD-4 and UCrCl | −31.95[*] [−40.53 to −27.26] | 51.26 [36.04 to 71.27] | −115.17 [−135.17 to −99.45] | 166.43 | 47.18% [37.21 to 64.84] |
| MDRD-6 and UCrCl | −20.68[*] [−27.30 to −16.93] | 44.28 [30-03 to 59.03] | −85.64 [−100.39 to −74.40] | 129.92 | 33.78% [27.53 to 39.90] |

**Notes.**

[*]$p < 0.0001$.

95% CI, 95% confidence interval; MDRD4, four-variable Modification of Diet in Renal Disease Study equation; MDRD6, six-variable Modification of Diet in Renal Disease Study equation; CKD-EPI, Chronic Kidney Disease Epidemiology Collaboration equation; CG, Cockroft–Gault formula; UCrCl, urinary creatinine clearance; APE, absolute percentage error.

ischemic stroke of the brainstem. Another was a patient with neurological degenerative disease who presented with bacterial pneumonia and respiratory failure.

## DISCUSSION

In this study, we found that GFR estimated with the four commonly used predictive equations was significantly higher than that obtained with the UCrCl method. Modest agreement was found between the predictive equations and UCrCl. Both the MDRD-4 and MDRD-6 formulas should not be used to determine GFR in the medical critically ill patients because of their large bias, wide precision, and high APE. In our cohort, the equation with the best performance for GFR estimation was the CG formula, followed by the CKD-EPI equation. However, the precision of both formulas remains a concern, due to the width of the limit of agreement. The poor performance of all the predictive equations was more evident when the UCrCl was <60 mL/min. However, the CG formula outperformed the others, particularly in the normal-UCrCl group, in which the bias was not significantly different from that of UCrCl with an acceptable APE. Based on our findings, the eGFR from commonly used predictive equations must be used with caution in critically ill patients, especially patients who develop acute kidney injury or have UCrCl <60 mL/min. We also found that the incidence of ARC in our study was low.

GFR is an index representing the filtration rate capability of the glomerulus, which is an accepted parameter for the assessment of global kidney function. GFR represents the amount of blood filtered per unit of time, but not necessarily kidney injury. GFR solely refers to the filtration of molecules by the glomeruli, without secretion and reabsorption by the renal tubules (*Seller-Perez et al., 2013*). Inulin is the substrate mainly used to measure

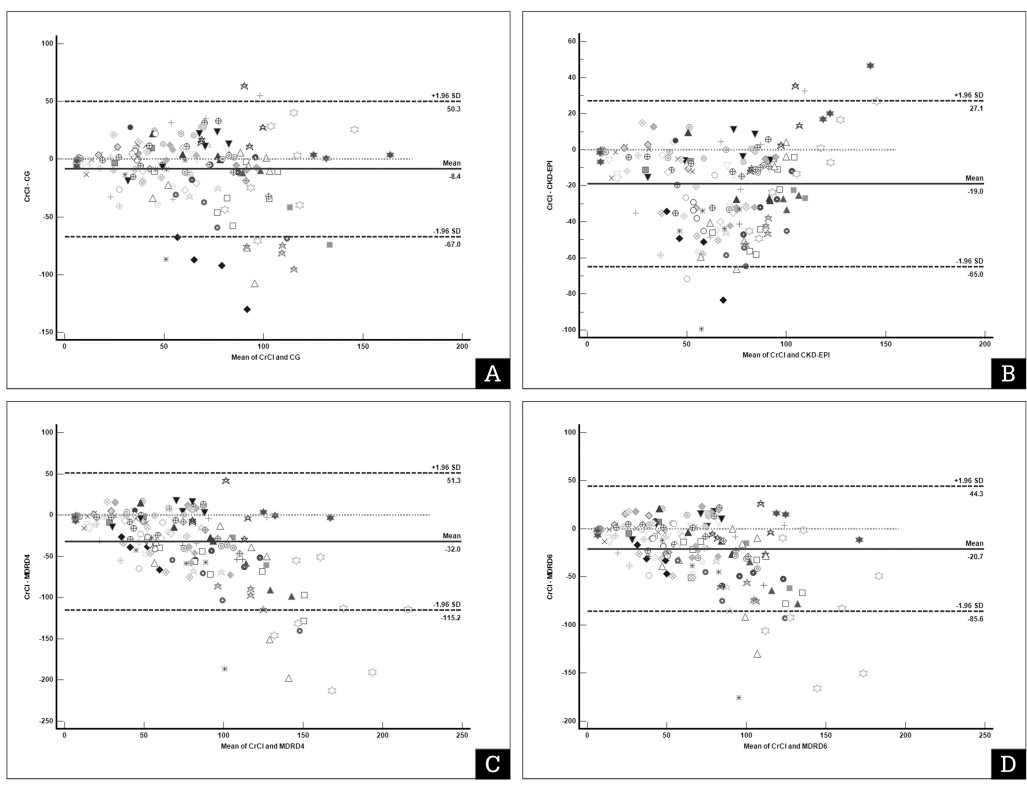

**Figure 1** **Bland–Altman plot with multiple measurements per subject of urine creatinine clearance (UCrCl) and estimated glomerular filtration rate (GFR) using predictive equations.** (A) Cockroft–Gault (CG) formula; (B) Chronic Kidney Disease Epidemiology Collaboration (CKD-EPI) equation; (C) Four-variable Modification of Diet in Renal Disease Study (MDRD4) equation; (D) Six-variable Modification of Diet in Renal Disease Study (MDRD6). Each marker represents a pair of UCrCl and estimated GFR. The x-axis represents the difference between the standard UCrCl calculation and the equation for estimating the glomerular filtration rate. The Y-axis represents the mean UCrCl and the equation for estimating the glomerular filtration rate. The solid line represents the bias (the mean difference obtained across the range of values), whereas the dashed lines represent the limits of agreement.

GFR, but its availability is limited (*Seller-Perez et al., 2013*). Iohexol solution, iothalomate and chromium-ethylenediaminetetraacetic acid (51Cr-EDTA) infusion are among the exogenous substance mainly use in the research setting rather than in the usual clinical practice due to its availability and cost (*Toffaletti, 2017*). Therefore, intrinsic substances, particularly serum creatinine, are widely used to determine kidney function (*Seller-Perez et al., 2013*). The standard UCrCl method requires 24 h of urine collection and a long turnaround time for laboratory results. Therefore, predictive equations for estimating GFR have been proposed and tested for accuracy and reliability in comparison with standard measurements.

Four commonly used equations, the CG, CKD-EPI, MDRD-4, and MDRD-6 formulas, have been widely applied in clinical practice (*Sunder et al., 2014*). Unfortunately, all these predictive equations were originally derived from outpatients or patients with specific conditions. Most predictive equations use simple parameters, such as age, sex, body weight,

**Table 5  Bias, precision, and absolute percentage error of estimated glomerular filtration rate using predictive equations versus urinary creatinine clearance (UCrCl) in low and normal UCrCl groups.**

| Pair agreement | Low UCrCl group (UCrCl <60 mL/min) (N = 98) | | | | Normal UCrCl group (UCrCl ≥ 60 mL/min) (N = 72) | | | |
|---|---|---|---|---|---|---|---|---|
| | UCrCl *vs* CG | UCrCl *vs* CKD-EPI | UCrCl *vs* MDRD-4 | UCrCl *vs* MDRD-6 | UCrCl *vs* CG | UCrCl *vs* CKD-EPI | UCrCl *vs* MDRD-4 | UCrCl *vs* MDRD-6 |
| Bias (mL/min) [95% CI] | −13.69[*] [−19.53 to −7.85] | −26.59[*] [−31.35 to −21.83] | −30.48[*] [−38.21 to −22.71] | −20.72[*] [−26.96 to −14.48] | −2.30[**] [−9.46 to 4.86] | −10.36[*] [−14.90 to −5.82] | −38.57[*] [−50.30 to −26.83] | −24.02[*] [−33.0 to −15.03] |
| **Precision** | | | | | | | | |
| Upper limit of agreement (mL/min) [95% CI] | 43.37 [33.36 to 53.38] | 19.93 [11.77 to 28.09] | 45.29 [32.01 to 58.58] | 40.25 [29.56 to 50.95] | 57.42 (45.12 to 69.71] | 27.50 [19.70 to 35.28] | 59.27 [39.13 to 79.41] | 50.95 [35.51 to 66.38] |
| Lower limit of agreement (mL/min) [95% CI] | −70.76 [−80.77 to −60.75] | −73.11 [−81.27 to −64.98] | −106.21 [−119.80 to −92.92] | −81.70 [−92.39 to −71.0] | −62.02 [−74.32 to −49.73] | −48.22 [−50.02 to −40.43] | −136.41 [−156.55 to −116.27] | −98.98 [−114.41 to −83.55] |
| Width of limit of agreement (mL/min) | 114.13 | 93.04 | 151.50 | 121.95 | 119.44 | 75.72 | 195.68 | 149.48 |
| APE (%) [95% CI] | 29.57 [21.81 to 43.39] | 73.26 [55.06 to 86.30] | 70.82 [44.33 to 98.76] | 47.84 [33.55 to 70.97] | 14.72 [10.87 to 23.80] | 13.77 [12.09 to 21.67] | 34.43 [20.10 to 47.48] | 22.60 [17.56 to 32.79] |

**Notes.**

[*] $p < 0.0001$.

[**] $p = 0.52$.

95% CI, 95% confidence interval; MDRD4, four-variable Modification of Diet in Renal Disease Study equation; MDRD6, six-variable Modification of Diet in Renal Disease Study equation; CKD-EPI, Chronic Kidney Disease Epidemiology Collaboration equation; CG, Cockroft–Gault formula; UCrCl, urinary creatinine clearance; APE, absolute percentage error.

and SCr to calculate GFR. The CG formula was initially established using 24-hour urine collection from hospitalized medical patients with normal kidney function. Therefore, its performance in critically ill patients or patients with kidney impairment remains inconclusive. The CKD-EPI formula was also generated from outpatients with mild kidney impairment and mostly non-surgical patients, and the two MDRD formulas were also derived from outpatients with mild to moderate kidney impairment (*Bragadottir, Redfors & Ricksten, 2013*). The validation of all the above predictive formulas for GFR estimation in critically ill patients is therefore limited.

Nonetheless, the standard UCrCl method in critically ill patients is sometimes difficult to perform because it requires 24 h of urine collection and cannot provide immediate results for decision-making when rapid drug administration is needed. Furthermore, several issues in critically ill patients may disturb urinary creatinine filtration, such as rapid hemodynamic changes, vasopressor and inotrope administration, and inconsistent urine flow during the resuscitation period (*Sunder et al., 2014*). *Bragadottir, Redfors & Ricksten (2013)* found that in postoperative cardiovascular patients who developed early acute kidney injury (AKI), the standard UCrCl method showed an unacceptable repeatability and large bias compared to the measurement of UCrCl using the chromium-ethylenediaminetetraacetic acid (51Cr-EDTA) infusion technique. However, our cohort included more hemodynamically stable patients who were admitted to the medical ICU or RCU, and only one (2%) patient required a low dose of vasopressor, with all patients receiving a urinary catheter for urine collection. In addition, the main causes of admission in our patients were respiratory problems, and the hemodynamic disturbance was lower than that in the previous study. Therefore, those confounding factors may not disturb the excretion and measurement of UCr. We then hypothesized that UCrCl could be used as the gold standard for GFR measurement in our study.

Based on the agreement test between UCrCl and eGFR using a predictive equation, our results are concordant with those of previous studies in which common predictive equations overestimated GFR compared to the UCrCl method (*Bragadottir, Redfors & Ricksten, 2013*; *Baptista et al., 2014*). A French study reported that eGFR estimated by the CG, CKD-EPI, and MDRD formulas tended to overestimate compared to UCrCl in patients with normal kidney function. It also should be noted that the correlation between UCrCl and eGFR from predictive equations in French study was poor and had a large bias and poor repeatability in either medical or surgical critically ill patients (*Ruiz et al., 2015*). One explanation for this disagreement is the possible occurrence of ARC, which is commonly observed in patients who are young, have experienced trauma, and have less severe disease (*Nei et al., 2020*). Several studies reported the incidence of ARC to range between 8% and 55%, which could possibly reflect the poor agreement between UCrCl and eGFR (*Baptista et al., 2020*; *Abdul Rahim & Md Ralib, 2018*). However, the incidence of ARC in our study was very low at 2%. Therefore, the abovementioned limitation of the predictive equations could stem from the calculation method using demographic data and Scr values, which are less sensitive to kidney impairment and cannot match the dynamic changes in kidney function in critically ill patients.

We also performed an agreement test between eGFR and UCrCl in the low- and normal-UCrCl groups. UCrCl <60 mL/min in the low-UCrCl group may reflect an early stage of AKI in critically ill patients, requiring dose adjustment for drug administration. Several reports in AKI noted that the eGFR from predictive equations, including the CG, CKD-EPI, and MDRD formulas, performed poorly compared to the actual measurement of GFR. The bias was large, and poor precision of eGFR was reported in several studies (*Tsai et al., 2018*; *Kirwan, Philips & Macphee, 2013*; *Candela-Toha et al., 2018*; *Carlier et al., 2015*). As a result, UCrCl was suggested in the critically ill patients with AKI, instead of eGFR formulas.

A recent secondary analysis in the PERMIT trial found that the eGFR from predictive equations performed poorly and tended to be overestimated when compared to the GFR of UCrCl method. However, they found that the best predictive equation with the lowest bias was the MDRD-6 equation (*Al-Dorzi et al., 2021*). Our results do not support this finding as we found a poor performance of MDRD-6 formula for estimating GFR, but the CG formula was the best predictive equation in our study. Both MDRD formulas had a larger bias and an unacceptable APE and should not be used in medical critically ill patients. We also found that the CG formula had the best performance in the normal-UCrCl group with an insignificantly biased and acceptable APE. The different types of patients recruited in our study and the PERMIT trial, which was majority surgical critically ill patients, could be the reason for this discordance. In addition, the CG formula was initially derived from patients who were admitted to the general medical ward without significant renal impairment. This advantage of the CG formula may explain our findings.

Our finding also confirmed that eGFR formulas overestimated standard GFR measured by UCrCl method. The overestimation of GFR may resulted in the overdose and complication of several medications that required renal excretion, for example antibiotics and chemotherapy (*Nakata et al., 2012*). The appropriated selection of eGFR formula that have the lowest bias may reduce those undesired complications. From our study, we prefer CG formula for estimating GFR to adjust the dose of medication in medical critically ill patients, particularly in normal kidney function. However, in the patient who developed acute kidney injury, that indicated by elevated SCr, the standard UCrCl method should be considered due to the high bias of all predictive equations. One should debate that this standard method requires times to perform and 24-hours urine collection may delay the appropriate medication dose adjustment. The recent study found that UCrCl with 1–2 h urine collection time strongly correlated to GFR by urine inulin clearance ($r = 0.92$) with the small bias of 11 mL/min in the patient who had GFR < 60 mL/min (*Carlier et al., 2015*). In addition, the shorter urine collection by 4 hours- and 8 hours-urine collection for UCrCl calculation was evidently equivalent to the standard 24-hours UCrCl method (*Cherry et al., 2002*; *da Silva et al., 2010*; *O'Connell et al., 1993*).

Although the measurement of SCys correlated better with UCrCl than Scr in patient with acute kidney injury (*Villa et al., 2005*), the correlation of eGFR estimated by SCys and SCr was uncertain (*Inker et al., 2021*; *Inker et al., 2012*). In addition, the inflammatory response and rapid inflammatory cell turn over in the medical critically ill patients will disturb the cystatin c production (*Randers & Erlandsen, 1999*). The recent consensus from

KDIGO (Kidney Disease Improving Global Outcomes) suggested to use creatinine-based equation to determine GFR in the acutely ill patients and cystatin-based equation is suitable for chronic kidney disease (*Andrassy, 2013*).

The results of our study should be applied considering its limitations and strengths. First, our study was a secondary analysis of the ongoing nutrition therapy trial, in which the inclusion criteria were specific and rigid. Most of the patients in the primary study were clinically stable and well resuscitated, and a very small number of patients required inotropes or vasopressors. Therefore, our results cannot be applied to more severe cases with hemodynamic instability or during the resuscitation period. Second, most patients in the original study were critically ill solely from a medical perspective; therefore, the application of our findings in surgical or trauma patients must be done with caution. In addition, the incidence of ARC in our patient cohort was very low compared to that in surgical, burn, and critically ill trauma patients; therefore, further study is warranted in surgical critical care patients. The number of patients in our study was relatively smaller than several previous studies. The larger population study may result in variable finding. Finally, patients in our study had normal kidney function throughout the study period. Thus, the interpretation of our results must be strictly limited to patients with normal kidney function; medical critically ill patients who developed early AKI could not be appropriately applied from our results. Although our study has several limitations, our results may remind physicians of the poor performance of frequently-used predictive equations to estimate GFR for dose adjustment of medications. In patients who require drugs with a narrow therapeutic range and are mainly excreted by the kidney, close monitoring of the peak blood drug level and dose-related adverse reactions is recommended, in addition to the standard determination of GFR.

## CONCLUSIONS

Although eGFR from commonly used predictive equations has limited performance in estimating UCrCl in critically ill medical patients, the CG formula demonstrated the best performance to estimate GFR against the standard UCrCl method in our study. The CG formula gave the lowest bias and acceptable APE in the medical critically ill patients, but the performance of this formula became weaker in the kidney impairment condition or UCrCl <60 mL/min. We prefer to use the eGFR formula with caution in the medical critically ill patients with acute kidney injury and suggest that the standard method for UCrCl should be considered in this clinical setting. However, our findings need to be confirmed by further larger studies involving critically ill patients.

## ACKNOWLEDGEMENTS

The authors would like to sincerely thank the research coordinators and research nurses at the Clinical Research Center (CRC) of the Faculty of Medicine at Prince of Songkla University for their contribution to patient recruitment.

### Funding

The authors received no funding for this work.

### Competing Interests

The authors declare there are no competing interests.

### Author Contributions

- Suwikran Wongpraphairot conceived and designed the experiments, analyzed the data, prepared figures and/or tables, authored or reviewed drafts of the article, and approved the final draft.
- Attamon Thongrueang conceived and designed the experiments, performed the experiments, prepared figures and/or tables, and approved the final draft.
- Rungsun Bhurayanontachai conceived and designed the experiments, performed the experiments, analyzed the data, prepared figures and/or tables, authored or reviewed drafts of the article, and approved the final draft.

### Human Ethics

The following information was supplied relating to ethical approvals (i.e., approving body and any reference numbers):

the human research ethics committee at the Faculty of Medicine, Prince of Songkla University (registration numbers REC.64-416-14-4)

### Data Availability

The raw data is available in the Supplementary File.

### Supplemental Information

Supplemental information for this article can be found online at http://dx.doi.org/10.7717/peerj.13556#supplemental-information.

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
