# Peer review of "Glomerular filtration rate correlation and agreement between common predictive equations and standard 24-hour urinary creatinine clearance in medical critically ill patients"

_PeerJ, doi:10.7717/peerj.13556_

## Round 0.1 · original submission · Major Revisions

I suggest focusing more on the discussions and conclusions on the own results. In addition, more details regarding the study methodology should be provided.

·

Basic reporting

Thank you for the invitation to review this manuscript. This is an interesting piece of work. However, there are several limitations in the manuscripts which I believe should be considered before making any decision on this submission.

1. The abstract section of the manuscript has more details on the study rationale and methods. The authors provided very limited results in this section. I suggest including more numerical results rather than descriptive statements i.e. correlation coefficients between equations, demographics of patients, etc.

2. The conclusion section in the abstract is quite general, and does not represent the main findings from the results. The conclusion should be according to the results of this study rather than a general statement. These statements can be adjusted after conclusive remarks.

3. Authors have well explained the rationale of this study but did not provide any information on the previous studies conducted on the same topic. It's obvious that the current study is not the first one in this field, authors need to provide the reference from previous studies and explain the justification that why this study is needed in the presence of other studies.

Experimental design

The methodology section is very brief and dispersed. Authors may follow the following headings to re-construct this section; ethics, study location, study population with inclusion and exclusion criteria, data collection, operational definitions, normal laboratory values used to classify morbid conditions in this study, primary outcome variables, and statistical analysis. I appreciate that the authors have presented most of this information but it would be more convenient for readers to understand the methods divided into various sections.
The sample size estimation must be provided under a separate heading with the name of the equation used for such estimation.
The level of correlation should be explained in the statistical analysis section. i.e. the cut-off values used in the classification of correlation coefficients
There is a lack of information on how baseline serum creatinine was estimated among patients. Since the baseline values are important for GFR estimation, the authors did not discuss the impact of over-and under-estimation of baseline Serum creatinine on the findings of this study.

Validity of the findings

The findings of this study are well presented in the form of tables and figures. However, the discussion section needs the attention of the authors. There is a need to discuss the impact of cystatin-based equations on the findings. Do authors provide a hypothesis that GFR based on the UCrCl method may be equal, higher, or lower than what is estimated with cystatin-based equations? I think the authors should also consider the low sample size as a potential limitation of this study. Since authors have carried out vigorous inferential tests on the data, a large sample size may result in variable findings. Please discuss the same in the limitation section.
Again, the conclusion section is quite general, authors should state the major results in this section followed by directions for future research and recommendations for practice.

Reviewer 2 ·

Basic reporting

Thanks for asking me to review this study. Wongpraphairot et al. attempted to compare the four commonly used equations for estimation of GFR among critically ill patients with respect to UCrCl. I agreed with the authors that these equations primarily originated from other populations without involving critically ill patients. There is a need to explain that in the presence of a large volume of studies (given below) where the use of these equations is discouraged, why this study is needed, and what new information is being produced through the analysis in this study.

https://pubmed.ncbi.nlm.nih.gov/34055406/

https://jintensivecare.biomedcentral.com/articles/10.1186/2052-0492-2-31

https://academic.oup.com/ndt/article/25/1/102/1909271

https://jamanetwork.com/journals/jamainternalmedicine/fullarticle/2731708

https://link.springer.com/article/10.1007/s40262-018-0636-7

https://journals.sagepub.com/doi/pdf/10.1177/0310057X1804600107

I agree that there is a need for more data and the authors may have used these grounds to conduct this study. But what new information is brought up through this study is not clear to me.

Experimental design

The methodology and study design are sound in this study. However, the method section requires more clarification, particularly the terms used throughout the manuscript. For example, low- and normal-UCrCl groups should be defined in the method section.
Since this study is conducted on a cohort from a previously initiated trial, there is a need to briefly describe the primary cohort from which patients` data were extracted.
There is no information on the confounding adjustment in the statistical analysis section, the CrCl and GFR are also confounded by various factors which the authors have already discussed in the discussion section. Were these factors adjusted in the correlation analysis?

Validity of the findings

Discussion section: Authors have quoted the statement "It also should be noted that the correlation between UCrCl and eGFR from predictive equations was poor and had a large bias and poor repeatability in several critical care settings, either for medical or surgical patients----------------------"

However, in their results, they explained that "We performed a correlation analysis for UCrCl and eGFR from four commonly used predictive 188 equations using the Spearman rank correlation coefficient. All eGFR values from the predictive 189 equations showed a good correlation. Nevertheless, the MDRD-6 equation had the strongest correlation with the standard UCrCl method, with a correlation coefficient of 0.76".... There is a need to explain the deviations

Additional comments

Can I assume that the authors are trying to convince the use UCrCl method instead of these four equations in critically ill patients?

Reviewer 3 ·

Basic reporting

The language should be clearer and there is still way to improve

Experimental design

1) The author should stress the basis of using 24-hour Urinary Creatinine Clearance in the setting of acute kidney injury in the introduction.
2) It is best to include patients' urine volume among these clinically ill patients and their baseline kidney problem or eGFR. Other vital signs are essential too.
3) The definition of a more stable patient admitted to ICU should be clearly defined as the definition of the targeted cohort is very important for the reliability of the results (In methodology).

Validity of the findings

1) Regarding the discussion, what other literature suggested the best formula for the critically ill patient? Compare authors finding with other published literature.

---

## Round 0.2 · accepted · Accept

The reviewers' concerns were resolved by the authors.

·

Basic reporting

I have no more concerns on this submission.

Experimental design

I have no more concerns on this submission.

Validity of the findings

I have no more concerns on this submission.

Additional comments

I have no more concerns on this submission.

Reviewer 2 ·

Basic reporting

Authors had addressed all the relevant comments and made necessary modifications in the manuscript clearly.

Experimental design

Authors had addressed all the relevant comments and made necessary modifications in the manuscript clearly. Although i still feel that current research does not fill knowledge gap as already so much work has been done on renal function estimating equations and urine creatinine clearance

Validity of the findings

In the revised manuscript, authors have added study rationale and significance.

Additional comments

Overall authors have addressed all the comments in the revised version of manuscript.